# Facile Splint-Free Circularization of ssDNA with T4 DNA Ligase by Redesigning the Linear Substrate to Form an Intramolecular Dynamic Nick

**DOI:** 10.3390/biom14081027

**Published:** 2024-08-18

**Authors:** Wenhua Sun, Kunling Hu, Mengqin Liu, Jian Luo, Ran An, Xingguo Liang

**Affiliations:** 1State Key Laboratory of Marine Food Processing & Safety Control, College of Food Science and Engineering, Ocean University of China, No. 1299 Sansha Road, Qingdao 266404, China; 2Laboratory for Marine Drugs and Bioproducts, Qingdao Marine Science and Technology Center, No. 1 Wenhai Road, Qingdao 266237, China

**Keywords:** ssDNA circularization, permuted sequence, T4 DNA ligase, splint-free ligation

## Abstract

The efficient preparation of single-stranded DNA (ssDNA) rings, as a macromolecular construction approach with topological features, has aroused much interest due to the ssDNA rings’ numerous applications in biotechnology and DNA nanotechnology. However, an extra splint is essential for enzymatic circularization, and by-products of multimers are usually present at high concentrations. Here, we proposed a simple and robust strategy using permuted precursor (linear ssDNA) for circularization by forming an intramolecular dynamic nick using a part of the linear ssDNA substrate itself as the template. After the simulation of the secondary structure for desired circular ssDNA, the linear ssDNA substrate is designed to have its ends on the duplex part (≥5 bp). By using this permuted substrate with 5′-phosphate, the splint-free circularization is simply carried out by T4 DNA ligase. Very interestingly, formation of only several base pairs (2–4) flanking the nick is enough for ligation, although they form only instantaneously under ligation conditions. More significantly, the 5-bp intramolecular duplex part commonly exists in genomes or functional DNA, demonstrating the high generality of our approach. Our findings are also helpful for understanding the mechanism of enzymatic DNA ligation from the viewpoint of substrate binding.

## 1. Introduction

Efficient preparation of highly pure single-stranded DNA (ssDNA) rings is essential due to the ssDNA rings’ wide variety of applications in biotechnology, such as DNA detection and the assembly of topologically constrained DNA constructs. For example, ssDNA rings are commonly used as the template for rolling circle amplification (RCA) to detecting nucleic acids from target biological species such as viruses and pathogenic bacteria [1,2]. Functional ssDNA rings have been widely utilized for drug delivery [3,4,5], for high sensitivity detection [6,7], and as nucleic acid medicine [8]. They can also be used for construction of molecular motors [9,10], DNA origami [11,12], DNA nanoflower [13,14], and left-handed DNA duplex [15] because a circular DNA possesses unique features in dynamics, topological constraints, and high resistance to exonucleases. However, preparation and purification of ssDNA rings, especially at high concentrations for short strands, remain significant challenges.

A splint is essential for the conventional enzymatic preparation of a circular ssDNA to draw two ends close by forming a nick (Figure 1A). However, two major issues have to be addressed: avoidance of by-products of polymers and removal of splints after circularization. In particular, when the substrate is shorter than 40 nt or at high concentrations (e.g., >10 μM), the polymerization is almost unavoidable. Although the linear splint can be digested by exonuclease, achieving 100% digestion is very difficult. To address these problems, several efforts have been made [16,17,18,19]. Petersen et al. utilized restriction enzymes to cleave two substrates to generate complementary sticky ends for circularization [20]. However, the prepared DNA ring had a very stable hairpin structure (with a stem longer than 20 bp) which may limit its applications. In addition, this approach cannot be used to circularize a linear DNA without a very stable secondary structure. Magnus Stougaard et al. introduced an extra hairpin (designated as the turtle structure) with a long stem (>20 bp) for splint-free circularization, but still only a special sequence could be circularized [21]. Furthermore, CircLigase (a DNA ligase) can directly connect the 5′ and 3′ ends of ssDNA [22,23], but some secondary structures can decrease the efficiency (e.g., when the two ends were impeded to come close, the circularization yield decreased to only 26% [22]), and the specially engineered CircLigase is relatively expensive. Ligation by CircLigase also requires a high concentration of enzyme and the assistance of Mn^2+^, which further limits its applications.

Recently, we established a novel splint-free approach for the circularization of ssRNA with T4 RNA ligase 2, in which an intramolecular pseudo-nick with mismatches forms when using the permuted linear RNA substrate [24]. On the other side, we found that DNA ligation with T4 DNA ligase (T4 Dnl) occurs when one side flanking the nick is only several base pairs long (a mini-hairpin) [25]. As early as 2005, Kuhn et al. reported that T4 Dnl can circularize ssDNA in the absence of any obvious secondary structure, although the efficiency was extremely low [26]. While it has been reported that ligation by T4 DNA ligase occurs only when two duplex parts longer than 5 bp (11 bp in total) flank the nick [27], the above studies inspired us to come up with an idea that ssDNA may be circularized by forming an intramolecular dynamic nick, similar to RNA circularization with T4 RNA ligase 2. In our present study, unexpectedly and interestingly, only 2–4 base pairs flanking the nick can be ligated by T4 Dnl. Based on this, we proposed a novel splint-free approach for facile ssDNA circularization with high generality. Once the desired circular ssDNA has a secondary structure containing five or more consecutive base pairs, the approach of permuted precursor (linear ssDNA) for circularization (PPC) can be used. With this method, DNA rings of various sizes and sequences can be easily prepared with a yield close to 100%.

## 2. Materials and Methods

### 2.1. Materials

All the DNA oligonucleotides (Appendix A) were purchased from Shanghai Sangon Biotech (Shanghai, China). T4 Dnl and its 10× buffer were purchased from Thermo Scientific (Pittsburgh, PA, USA). Prior to the circularization experiments, a phosphate residue was enzymatically introduced to the 5′ position of these linear DNAs by using T4 Polynucleotide Kinase (Thermo Scientic, Pittsburgh, PA, USA). Exonuclease I and Exonuclease III were obtained from AB clonal (Wuhan, China). All other chemicals were obtained from Sigma-Aldrich (St. Louis, MO, USA).

### 2.2. Circularization of Linear ssDNA

A typical circularization system (10 μL) contains linear DNA (3 or 10 μM), and T4 Dnl (2.5 U) in 0.1× T4 ligase buffer (4 mM Tris-HCl, 1 mM MgCl_2_, 1 mM DTT, 0.05 mM ATP, pH 7.8 at 25 °C). In most cases, the ligation was performed at 25 °C and terminated by incubating the mixture at 65 °C for 10 min. The ssDNA rings (usually diluted by 10 times of buffer containing 0.5× Exonuclease I buffer and 0.5× Exonuclease III buffer) were confirmed by digestion using Exonucleases I (1.0 U/μL) and Exonucleases III (1.0 U/μL) at 37 °C for 2 h. The products were subjected to 12% polyacrylamide gel electrophoresis (12% PAGE).

### 2.3. Evaluation of Circularization Yield and Selectivity

The data were analyzed by Image Lab software for PC Version 3.0 to quantify the fluorescence emission for each band. The calculating methods of yield and selectivity were the same as reported previously [28], using the following equations.
Yield (%) = C/(B + C + R) × 100%(1)
Selectivity (%) = C/(B + C) × 100%(2)
where C, B, and R were band intensities of desired circular ssDNA, by-products, and remaining linear DNA substrates, respectively.

### 2.4. Mfold Calculation

Secondary structures of DNA were simulated by the Mfold Web Server (http://www.unafold.org/) using “DNA folding form” [29]. The typical conditions are: the DNA sequence is circular, folding temperature = 25 °C, ionic conditions = 0 mM Na^+^, 1 mM Mg^2+^, and no extra constraint information.

## 3. Results

### 3.1. Unexpected Ligation of an Intramolecular Nick Flanking Only Two orThree Base Pairs to Circularize a 39-nt ssDNA

In our previous study, a 39-nt linear ssDNA (L0) containing the catalytic active center of 10-23 DNAzyme was used to develop a circularization approach by using an abnormally low concentration of salts and dropwise addition [19]. Here, we try to realize splint-free circularization by utilizing secondary structures of permuted linear ssDNA substrates (Figure 1A). The secondary structure of the desired circular ssDNA (C0) (Figure 1B) has five consecutive base pairs. As shown in Figure 1C, by permuting new ends on the five consecutive base pairs (L1), a dynamic intramolecular nick may form. The completely complementary parts flanking the nick are 2 bp and 3 bp, respectively, while others contain mismatches. Under normal ligation conditions (e.g., >20 °C), obviously this structure is difficult to form (*T*_m_ lower than 10 °C, Appendix A). Surprisingly, after being circularized by T4 Dnl at 25 °C for 24 h (Figure 1D), almost 100% of L1 (10 μM) converted to C1, the circular form (exactly the same as C0). This demonstrates that only several base pairs (2, or 3) flanking the nick are enough for ligation, although they form only instantaneously. It has been reported that T4 Dnl can bind about 18-bp-long duplex involving a nick in the middle [30]. We believe that T4 Dnl helps stabilize the dynamic structure and achieve circularization, although only a part of T4 Dnl’s binding surface contributes to binding.

It is noteworthy that no extra splint was added in this PPC approach. When using the previous method with a splint, the yield is only 45.8%, and approximately 54.2% of L0 is converted to circular dimer (0.1× buffer, Figure 1E). If a higher concentration of buffer (e.g., 1×) is used, more polymers should appear [19]. Moreover, by using PPC, L1 at high concentrations (30 μM, 50 μM, or 100 μM) is circularized in 0.1× T4 ligase buffer at 30 °C for 72 h (Figure 1F). For 30 μM and 50 μM, a yield of higher than 95% for C1 is obtained. Even for 100 μM, the yield is as high as 71.9%, and only 5.8% of the circular dimer (selectivity is 92.5%) is produced (Lane 2, Figure 1F). Obviously, compared to the splint-assisted method, the yield and selectivity of our new approach are relatively high due to intramolecular hybridization. One of the advantages is that there are almost no by-products of polymers, even when the substrate concentration is as high as 50 μM (Figure 1F).

### 3.2. The Length Limit for Efficient PPC

To assess the generality of this PPC approach, we tried to circularize ssDNA with various stem sequences (see sequence L2–L10 in Appendix A) of different lengths (3 or 4 bp at 3′-side, 2–4 bp at 5′-side). Interestingly, for most sequences, almost 100% yield can be obtained when ligation time is long enough (2–12 h). Unexpectedly, higher GC content of the 3′-side hairpin shows slower circularization. The rigid structure of the G-C pair may impede the ligation, which requires an allosteric change for the ligase [30,31]. When the stem is 2 bp long at the 3′-side (GAA loop), the ligation hardly occurs (Appendix A).

To further investigate the length limit of the linear substrate for ligation, as short as 16–20 nt ssDNAs were used. As shown in Figure 2A, the 3′-side of the sequences is fixed as a relatively stable hairpin (with a 3-bp stem and a GAA loop) [32,33], and the 5′-sides are different in lengths. Surprisingly, even for L11 (16 nt), its circular product can be obtained, although the yield is low (12.8%). Another band with obvious low mobility appeared (63.2%), assigned as the adenylated intermediate (AppDNA). With the length increase at the 5′-side, the circularization yield increases gradually (Figure 2B). For L15 (20 nt), a yield of 91.7% is obtained. When all three base pairs of the stem are A-T pairs (see sequence L16–L20 in Appendix A), the circularization yield decreases greatly, demonstrating that the 3′-side hairpin has to be stable enough for efficient circularization. These results demonstrate that T4 Dnl can ligate a nick with only a 5-bp-long DNA duplex part for ligase binding, which is much shorter than what Ng et al. showed [27]. The result for L11 also indicates that adenylation may have a shorter length limit than the ligation step [25].

### 3.3. Effect of Nick Location on Circularization

Design of the proper nick position is essential for PPC. Taking the target ssDNA ring C1 as an example, it contains five consecutive base pairs, and the nick can be designed at various positions within consecutive base pairs. As shown in Figure 3A, the nicks at various positions are named as n1–n8. Two different ssDNA concentrations were used for circularization in Figure 3B (0.6 μM) and 3C (10 μM). In Figure 3B, for n2, n3, and n4, circularization yield was 79.0%, 100%, and 75.1%, respectively. For n1 and n5–n8, no circularization product was observed, but 100%, 56.5%, 52.6%, 58.8%, and 86.9% of AppDNA were obtained, respectively. Even when extending the ligation time from 18 h to 36 h (10 μM substrate), no ring C1 was observed for n1 and n5–n8. We attempted to circularize the AppDNA of n1 further by using T4 DNA ligase in the absence of ATP, but no ssDNA ring was generated (Appendix A), indicating that this structure affects the third step for ligation greatly, but is not so sensitive for adenylation [30,34].

In addition, there is a significant difference in the circularization results between group n1–n4 and group n5–n8 (e.g., comparing n3 with n7). This may be related to the chirality (asymmetry) and binding preference of T4 Dnl. When we just changed the sequence direction (reverse 5′→3′ to 3′→5′ without changing the bases, Appendix A), the yield of the ssDNA ring increased from 0% (n7) to 40.7% (n7′) for the same position (Appendix A). It is noteworthy that the stem for 3′-side hairpin contains 3 bp (with one G-C pair) for both n3 and n7′. This may also be the reason why n8 produces by-products while n4 does not (Figure 3C). Accordingly, if the consecutive complementary region is short (only 5–6 bp), it is recommended to place the smaller loop at the 3′-side of the nick to ensure forming a relatively stable hairpin (with a stem ≥3 bp), which is consistent with that previously described in Section 3.2.

### 3.4. Optimize Reaction Conditions to Increase the Yield of Circularization by Decreasing or Avoiding Adenylation

As the short dynamic hairpins used in our new approach are not very stable, adenylation often occurs and should be avoided to achieve a high yield of circularization. To avoid the production of AppDNA, we optimized the reaction conditions. As shown in Figure 4A, we used L21 as the substrate to check the effects of buffer, temperature, and reaction time on adenylation. Besides forming the nick for ligation, L21 can also form a relatively stable hairpin with seven consecutive base pairs (Figure 4A). It is helpful to investigate the effects of other structures on both adenylation and circularization. As shown in Figure 4B, only AppDNA is observed as the product under normal buffer conditions (1× T4 ligase buffer, Lane 4). Considering that a higher ATP concentration favors the production of AppDNA [34], we tried diluted buffer (Figure 4B). When the 0.1× buffer was used, interestingly, 51.5% of circular DNA (C21) was observed (Lane 3). Even when the buffer condition decreased to 0.01×, 41.8% C21 was still obtained, and the left was AppDNA (Lane 2). Ligation temperature was also investigated, and it was found that relatively higher temperatures (25–37 °C) are favorable for obtaining C21, and most products are AppDNA at 16 °C (Figure 4C). We further investigated the effect of reaction time (Figure 4D). Unexpectedly, about 90% of L21 changed to AppDNA after 2 h, and 51.6% of C21 was obtained from AppDNA. This result indicates that the circularization in this case did not occur according to the normal ligation mechanism, in which free ligase (not attaching AMP) carries out ligation directly after adenylation without dissociation [30,31]. Accordingly, besides designing a smaller loop at 3′-side of the nick to form a relatively stable hairpin (with a stem ≥3 bp), lower ATP is required to give a high-yield for circularization.

### 3.5. Universality of the Method

Finally, we tried to utilize our PPC strategy to circularize several ssDNAs, namely L22–L26, with various sequences and lengths (22–62 nt, Figure 5). Among them, L22 is the complementary DNA of miRNA-127-5p (regulating matrix metalloproteinase 13 expression and interleukin-1 beta-induced catabolic effects in human chondrocytes) [35]; L23 is a part of the bacteriophage M13 (origin of replication); L24 is an oligoDNA complementary to the dimer of miRNA-21 that serves as an important indicator for human health in gut microbiota and nonalcoholic fatty liver [36,37]; L25 is an RCA template that has been prepared using a splint-assisted method for producing functional DNA nanomaterials [38]; L26 is another RCA template which has been prepared by CircLigase I for nanopore fingerprint immunoassay [39]. As shown in Figure 5, for all these sequences, almost 100% of circularization yields are obtained by newly designed linear substrates. It is noteworthy that the concentrations of L23–L26 are as high as 10 μM (0.6 μM for L22). When 10 μM of L22 is used, most ligation products are circular dimers, probably due to the formation of a dimer hybrid (Appendix A).

## 4. Discussion

In vivo, DNA ligases mainly play a role in DNA repair and replication [40,41]. Due to the fact that DNA is synthesized by polymerase with high fidelity [42] and a nick is formed between two Okazaki fragments [43], mismatches are rarely produced flanking a nick in a cell. However, mismatches at the nick greatly affect the ligation [44,45]. It can be imagined that the specificity (fidelity) may not come from evolutionary pressure to distinguish the mismatched base pair flanking a nick from a fully matched one. On the other hand, the high efficiency requirement for sealing fully matched nicks may help evolve ligases which cannot ligate the nick flanked by a mismatched pair. It is well known that there are big differences in structure between mismatches and full matches. Certainly, once the nick is sealed when a mismatch is present, the “mutation” will be introduced, which is not conducive to the stability of the gene. However, it is rare for a ligase to encounter a mismatch during ligation in vivo, and DNA ligase cannot distinguish mismatches very well.

Paradoxically, we found that the ligation by T4 Dnl occurs efficiently when there are only 2–4 completely complementary base pairs on both sides of the nick (with other base pairs being mismatched). The reasonable explanation is that the ligation is efficient once the perfect nick structure (about 5 bp long) at the active site of T4 Dnl is formed. The ligase does not distinguish the structures well 2–4 bp away from the nick. In other words, apart from requiring complementary sequences just adjacent to the nick, T4 Dnl has lower requirements for complementarity for binding. It has been reported that ligation by T4 Dnl requires 5–6 completely complementary base pairs (10–11 bp in total) on both sides [27]. This result may come from the fact that they used chemically modified linkers that T4 Dnl could not bind well. The mismatched base pairs far from the nick may also bind T4 Dnl in our cases. Certainly, the more complementary base pairs are at the binding region, the more efficient the ligation is.

Our data shows that the ligation can be carried out when a dynamic (unstable) nick structure is formed (Figure 2 and Figure 3). A possible mechanism is proposed where the ligase can bind to a short hairpin at the 3′-side and wait for the 5′-end to bind. Without the ligase, the hybridization of the 5′-end to form the nick is so difficult that only two consecutive base pairs are present. This is further supported by the result that a mismatch at the second adjacent site is allowed, once there are several complementary base pairs on both sides of the nick (Appendix A).

In many cases, the adenylated intermediates are observed (Figure 2, Figure 3 and Figure 4). It can be explained that the dynamic nick structure is easy to break, so that the ligase easily falls off after adenylation, and the subsequent ligation is difficult at high ATP concentrations [31,46] because the concentration of free ligase (not attaching AMP) is too low for further ligation. At lower ATP concentrations, more free ligases are present so that ligation proceeds. Accordingly, the 0.1× ligase buffer (0.05 mM ATP) significantly favors circularization for PPC at 25–37 °C (Figure 4B). In these cases, the AppDNA can even be ligated to circular DNA in the presence of free ligase (not attaching AMP) at relatively higher concentrations. At this condition, it is also favorable to avoid intermolecular ligation due to a lower Mg^2+^ concentration (1 mM) [19]. Certainly, a high concentration of Mg^2+^ (but low ATP) can be used if the ligation efficiency is too low (Appendix A). Moreover, based on the ligation mechanism of T4 Dnl [30], the production of AppDNA is mainly influenced by ATP, rather than other components in the diluted buffer.

However, in some cases, the ligation will stop at adenylation. For example, n1 has no target ssDNA rings, only AppDNA (as shown in Figure 3), the ligation is inhibited even when ATP is removed after adenylation (Appendix A). The possible reason is that the structure of n1 significantly affects the third step of ligation. It has been found that T4 Dnl completes the final step of ligation with a significant allosteric effect [30,31]. Thereby, combining the above content, it is recommended to place the smaller loop at the 3′-side of the nick to ensure the formation of a relatively stable hairpin (with a stem ≥3 bp), and using a 0.1× ligase buffer significantly favors PPC.

We all know that circularizing short sequences (<40 nt) is a significant challenge. However, our method has notable advantages in this area. Although circularization can be achieved through the frozen/lyophilization/cyclization (FLC) method [17] and the “step-by-step“ method [19] for short sequences (<40 nt), the yields of these methods are only about 60%, compared to the almost 100% yield of our approach. Additionally, our approach does not require splint.

Almost all ssDNA has secondary structures involving complementary base pairs [18], which can be utilized in the PPC approach to prepare circular DNA. If there is only one hairpin structure, it is easy to design the ends within its stem; if there are two or more hairpins, one of the most stable stems with a small loop can be selected to design the ends (Figure 5). Therefore, the PPC method has strong generality for most sequences with relatively strong secondary structure. Because no splint is required, and the circularization yield is close to 100%, the ssDNA rings prepared by this strategy can be used as materials for detection analysis as well as producing corresponding nucleic acid medicines and other DNA biotechnology.

## 5. Conclusions

In conclusion, we have established a novel splint-free circularization approach with high efficiency, which can avoid polymerization by-products. The core of this method is to utilize the consecutive complementary regions of the DNA substrate itself to form a dynamic nick by permuting its ends. Although the stability of the nick affects the efficiency of circularization, high-yield ligation can be obtained when only 2–4 base pairs are present flanking the nick, which is not so difficult for many DNA sequences. Therefore, our PPC approach has wide applications in DNA biotechnology.

## Figures and Tables

**Figure 1 biomolecules-14-01027-f001:**
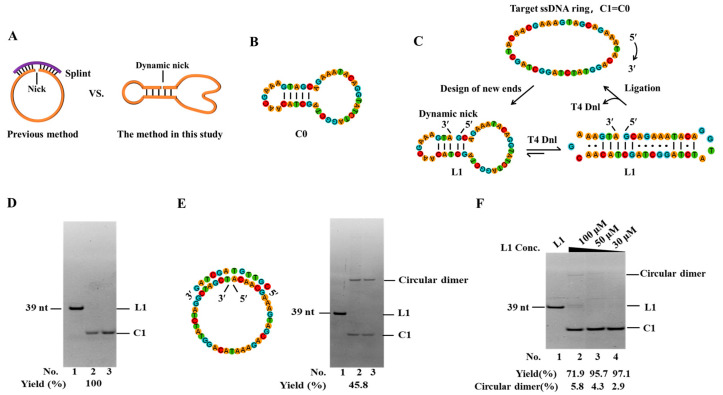
Efficient splint-free circularization using the permuted precursor for circularization (PPC) approach. (**A**) Schematic illustration of two circularization methods. (**B**) Secondary structure of C0 (simulated by Mfold). (**C**) Schematic diagram of circularizing L1 by PPC. The mismatched base pairs are also shown considering T4 Dnl can bind about 9-bp duplex at each side of the nick. (**D**,**E**) Electrophoresis analysis (12% PAGE) for circularization to prepare C1 by PPC and a previous approach, respectively. Lane 1, L1 (no T4 Dnl). Lane 2, L1 treated with T4 Dnl. Lane 3, the products in Lane 2 were further treated with Exonuclease I and Exonuclease III to remove linear substrate and polymers. Conditions: 10 μM linear ssDNA (20 μM splint in (**E**)), and 0.5 U/μL T4 Dnl, 0.1× T4 ligase buffer, 25 °C, 12 h. (**F**) Circularization of L1 at high concentrations. Other conditions: 100 μM L1, 1.0 U/μL T4 Dnl, 0.1× buffer, 30 °C, 72 h. Analyzed by 12% PAGE. Original images can be found in Appendix A.

**Figure 2 biomolecules-14-01027-f002:**
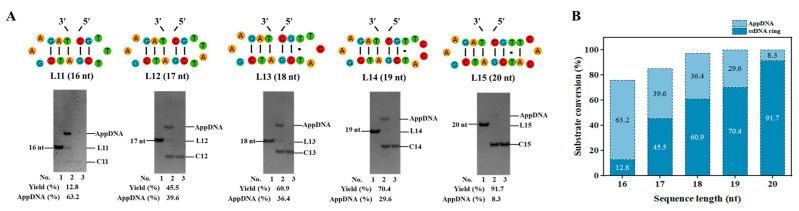
The length limit for efficient circularization. (**A**) Sequences with different lengths at 5′-side and electrophoresis results. Lane 1, linear substrate. Lane 2, ligation by T4 Dnl. Lane 3, the products in Lane 2 were treated with Exonuclease I and Exonuclease III. Original images can be found in Appendix A. (**B**) Circularization and adenylation yields for various sequences in Figure A. Conditions: 0.6 μM ssDNA, and 0.25 U/μL T4 Dnl in 0.1× T4 ligase buffer at 25 °C for 18 h, analyzed by 12% PAGE.

**Figure 3 biomolecules-14-01027-f003:**
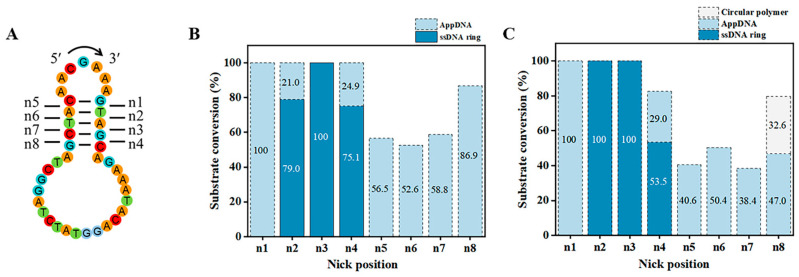
Effect of nick position on circularization. (**A**) Designed ends in different positions. (**B**,**C**) Circularization results of different nicks at two conditions. The same conditions for (**B**,**C**): 0.1× T4 ligase buffer, 25 °C. Different conditions for (**B**): 0.6 μM ssDNA, 0.25 U/μL T4 Dnl, 18 h; different conditions for (**C**): 10 μM ssDNA, 1.0 U/μL T4 Dnl, 36 h. Analyzed by 12% PAGE.

**Figure 4 biomolecules-14-01027-f004:**
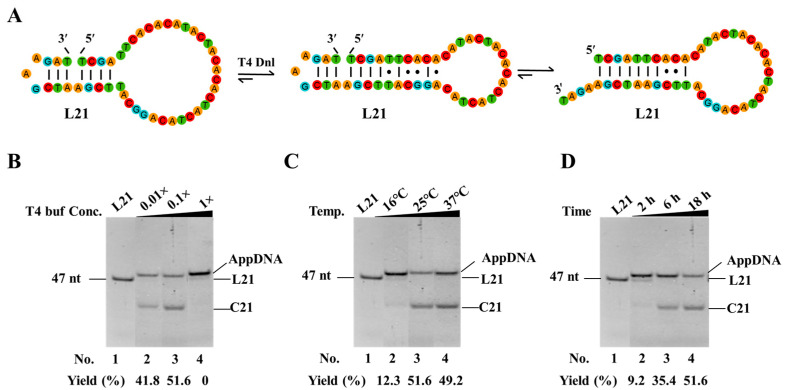
Effect of reaction conditions on circularization and adenylation of linear DNA substrates. (**A**) Possible secondary structures of L21. (**B**) Concentration of T4 ligase buffer. Other conditions: 0.6 μM L21, 0.15 U/μL T4 Dnl, 25 °C, 18 h. 1× T4 ligase buffer contains 10 mM MgCl_2_, 0.5 mM ATP, 10 mM DTT and 40 mM Tris-HCl. (**C**) Temperature. Other conditions: 0.6 μM L21, 0.15 U/μL T4 Dnl, 0.1× buffer, 18 h. (**D**) Ligation time. Other conditions: 0.6 μM L21, 0.15 U/μL T4 Dnl, 0.1× buffer. Original images can be found in Appendix A.

**Figure 5 biomolecules-14-01027-f005:**
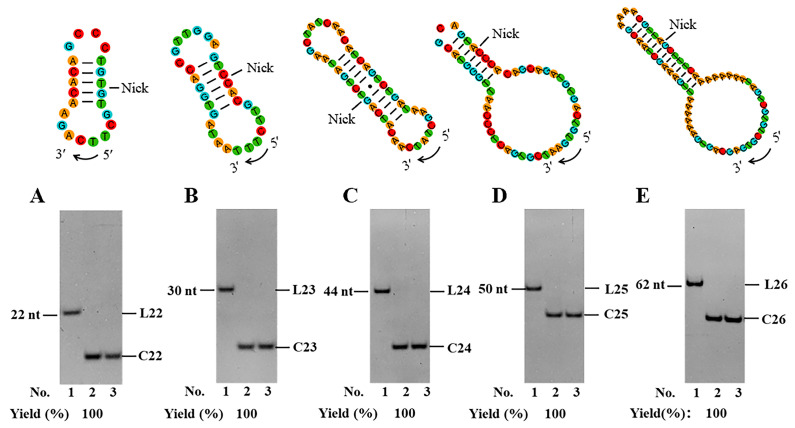
Application of the PPC method for circularization of various linear DNA substrates of L22 (**A**), L23 (**B**), L24 (**C**), L25 (**D**), and L26 (**E**). Lane 1, linear substrate; Lane 2, linear substrate treated with T4 Dnl; Lane 3, the products in lane 2 were digested with Exonuclease I and Exonuclease III. Ligation conditions: 0.6 μM linear DNA for L22 and 10 μM for other substrates, 0.5 U/μL T4 Dnl, 0.1× buffer, 25 °C, 24 h. Analyzed by 12% PAGE. Original images can be found in Appendix A.

## Data Availability

The data presented in this study are available on request from the corresponding author.

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
