# Peer review of "Facile Splint-Free Circularization of ssDNA with T4 DNA Ligase by Redesigning the Linear Substrate to Form an Intramolecular Dynamic Nick"

_biomolecules, 2024, doi:10.3390/biom14081027_

Round 1

Reviewer 1 Report

Comments and Suggestions for Authors

In this work, the authors developed a facile, splint-free circularization strategy using linear ssDNA that forms intramolecular dynamic nick to prepare circular ssDNA. The results showed high effectiveness and generality of the methods, with the high yield and selectivity of the circularization, and the mechanism of the method was well discussed. Henceforth, publication of this manuscript is recommended with minor revisions, as listed below.

1)        In the introduction and the discussion part, a more quantitative description of previous reports and comparison with this work are suggested, to present the novelty better. For example, the polymerization ratio, the concentration of ligase, circularization yield, etc. 

2)        In the Method section, all setup parameters for the secondary structures simulation by Mfold should be listed for clarity.

3)        When discussing the production of AppDNA by ATP using diluted buffer, is other components other than ATP in the buffer also diluted? Will those components influence the production of AppDNA?

4)        Some author information is missing.

Reviewer 2 Report

Comments and Suggestions for Authors

Excellent work! This manuscript develops a methodology to prepare small single-stranded DNA (ssDNA) circles without sacrificial splint strands. As the authors correctly pointed out, ssDNA circles are of great interest for their potential applications in biomedical research and materials science. Preparation of such ssDNA circles requires low DNA concentrations and requires additional splint strands, which have to be removed after synthesis. It is highly desirable to prepare ssDNA circles in high DNA concentrations without addition of splint strands. This work nicely addresses this challenge. The authors found that if the linear ssDNAs could form a dumbbell structures (with only totally 5-base pair pseudo-continuous duplex), they could be efficiently ligated into circles. Various experimental factors have been investigated for the optimal circlization. Excellent work! It will likely generate great impact on nucleic acid chemistry/technology. This reviewer highly recommends for publication.

Minor comments:

-          Ligation was conducted with 1 mM Mg2+. It would make more sense to use Mfold to fold the DNA at 1 mM Mg2+, instead of 10 mM Mg2+.

-          Fig. 2, AppDNA normally migrate only slightly slower than the original DNA strand (~ one base difference). Why the mobility difference is so large here?

-          Fig. 3A, the nick position is important. please adjust the lines to indicate the exact locations. Currently, they are misaligned for n3, n4, n7, n8.

-          Please clearly indicate the experimental difference between Fig. 3b and 3c.

-          On page 6, it states “that a higher ATP concentration favors the production of AppDNA”. Is there any reference to support this statement?

-          In Table S1, all DNA strands are listed. Are they all phosphorylated? Please explicitly stated since phosphorylation is required for ligation.
